# Structural basis for membrane tethering by a bacterial dynamin-like pair

Jiwei Liu[1], Jeffrey K. Noel[2] & Harry H. Low[1]

Dynamin-like proteins (DLPs) are large GTPases that restructure membrane. DLPs such as the mitofusins form heterotypic oligomers between isoform pairs that bridge and fuse opposing membranes. In bacteria, heterotypic oligomerisation may also be important for membrane remodelling as most DLP genes are paired within operons. How DLPs tether opposing membranes is unknown. Here we show the crystal structure of a DLP heterotypic pair from the pathogen *Campylobacter jejuni*. A 2:2 stoichiometric tetramer is observed where heterodimers, conjoined by a random coil linker, assemble back-to-back to form a tripartite DLP chain with extreme flexibility. In vitro, tetramerisation triggers GTPase activity and induces lipid binding. Liposomes are readily tethered and form tubes at high tetramer concentration. Our results provide a direct mechanism for the long-range binding and bridging of opposing membranes by a bacterial DLP pair. They also provide broad mechanistic and structural insights that are relevant to other heterotypic DLP complexes.

[1] Department of Life Sciences, Imperial College, London SW7 2AZ, UK. [2] Max Delbrück Center for Molecular Medicine, Kristallographie, Robert-Rössle-Strasse 10, Berlin 13125, Germany. Correspondence and requests for materials should be addressed to H.H.L. (email: h.low@imperial.ac.uk)

Classical dynamins and dynamin-like proteins (DLPs) are an expansive family of mechano-chemical GTPases that induce membrane fission and fusion. In eukaryotes, their functional repertoire includes the scission of vesicles in clathrin-mediated endocytosis, mitochondrial membrane maintenance, and viral surveillance and sequestration[1]. In bacteria, the role of DLPs is poorly understood although a high level of functional diversification is emerging. During DNA replication, the *Escherichia coli* DLP CrfC ensures equipartitioning of the chromosome by coupling nascent DNA strands[2]. DynA and DynB in *Streptomyces* form part of the divisome super-assembly that implements cell division during sporulation[3]. LeoA in enterotoxigenic *E. coli* (ETEC) has a tentative role in heat-labile toxin release via outer membrane vesiculation[4]. While the DLP DynA in *Bacillus subtilis* forms foci at the inner membrane in response to environmental stress conditions[5,6]. DynA has in vitro membrane fusogenic activity, and therefore may act as a molecular suture to repair lipid bilayer. Such a phenotype is compatible with the *Mycobacterium tuberculosis* DLP IniA, which confers drug tolerance to the antibiotics isoniazid and ethambutol[7]. Common to all these bacterial DLPs, with CrfC the exception, is a known or speculated requirement for heterotypic complex formation for functional integrity. Indeed, most bacterial DLPs exist as side by side pairs within operons[5,8]. *Streptomyces* DynA and DynB interact together in vivo, while *B. subtilis* DynA is comprised of two DLPs genetically fused into a single unit. The presence of up to 8 putative DLPs in some cyanobacteria further suggests the potential for multicomponent heterotypic supercomplex

formation, although only BDLP1 from *Nostoc punctiforme*[9] has so far been characterised with a speculative role in the fusion of photosynthetic membranes.

The emerging significance of heterotypic DLP association in bacteria is reminiscent of mammalian mitofusin and OPA1 DLPs, which form homotypic and heterotypic oligomers between isoform pairs that drive mitochondrial membrane fusion. These families are increasingly linked to neurological dysfunction and neurodegeneration[10–12]. Mitofusin 1 and 2 are located on the mitochondrial surface where both are essential for membrane maintenance. The precise role for each mitofusin remains unclear with evidence for both distinct[13] and cooperative membrane tethering and merging activity[11,14]. At the mitochondrial inner membrane, the OPA1 family form heterotypic complexes essential for cristae remodelling and fusion[15]. Long and short isoforms generate membrane anchored and soluble forms in the intermembrane space that complex together yet maintain distinct non-redundant roles[16,17].

DLPs typically comprise 3 conserved structural motifs as described for BDLP1 from the cyanobacteria *Nostoc punctiforme*[9]. These include the GTPase domain (G-domain) and 4-helix neck and trunk domains. The neck and trunk correspond to the bundle signalling element (BSE) and stalk in eukaryotic systems, with HD1 and HD2 domain terminology used for the mitofusins (Fig. 1a). At the trunk tip is a lipid-binding domain that is modified for topological and functional tuning within the cell. In BDLP1, the lipid binding domain constitutes a paddle, which provides transient membrane attachment based on nucleotide

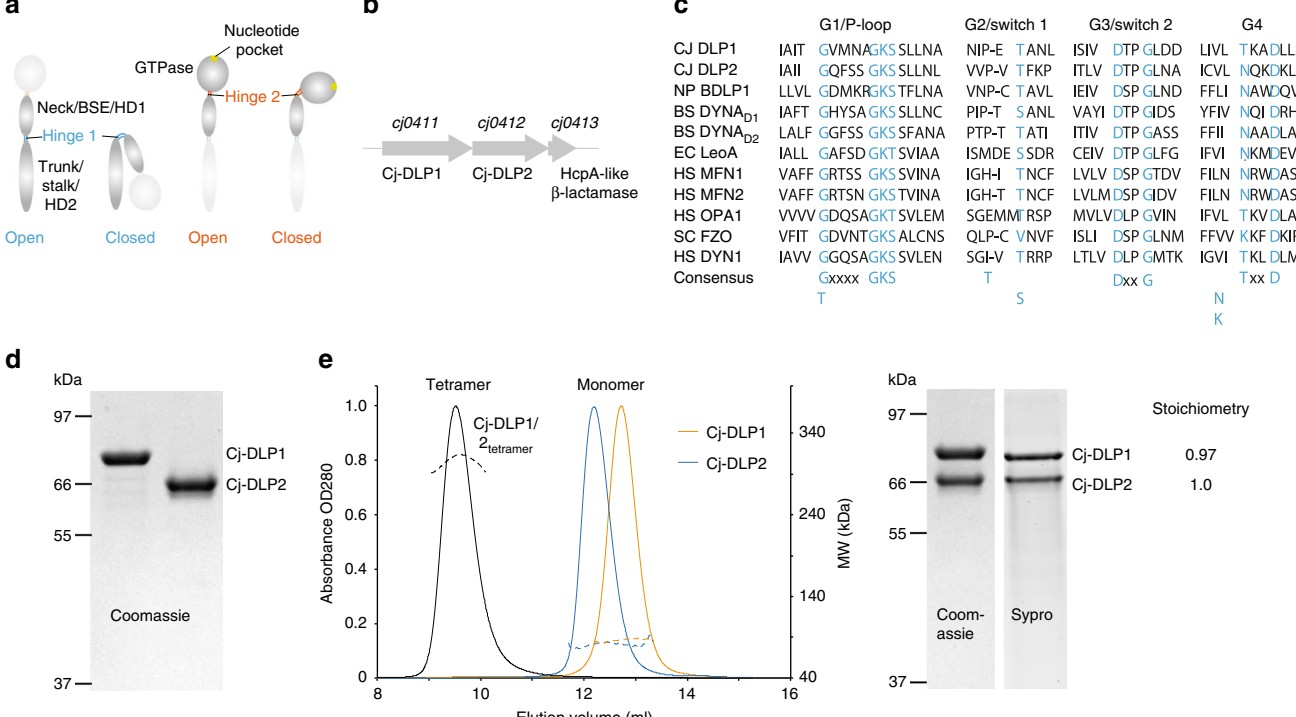

**Fig. 1** Biochemical purification and characterisation of Cj-DLP1 and Cj-DLP2. **a** Cartoon schematic providing nomenclature for dynamin family hinge 1 and hinge 2-mediated conformations. Many dynamin family members undergo large-scale conformational changes mediated by hinges 1 and 2 at the interface of trunk/neck or neck/G-domains, respectively. **b** *cj-dlp1* and *cj-dlp2* exist as a side-by-side gene pair within an operon that putatively includes an HcpA-like β-lactamase. **c** Alignment of Cj-DLP1 and Cj-DLP2 G1-G4 GTP binding motifs with other members of the dynamin family. Cj-DLP1 (Uniprot accession CJ0411), Cj-DLP2 (CJ0412), *Nostoc punctiforme* BDLP1 (B2IZD3), *Bacillus subtilis* (P54159), *Escherichia coli* LeoA (E3PN25), human Mitofusin 1 (Q8IWA4), human Mitofusin 2 (O95140), human OPA1 (O60313), *Saccharomyces cerevisiae* Fzo1p (P38297), human Dynamin 1 (Q05193). Consensus sequence is highlighted in blue. **d** Cj-DLP1 and Cj-DLP2 purify as 84.5 kDa and 71.2 kDa proteins, respectively, as shown by SDS–PAGE. **e** SEC-MALS shows that Cj-DLP1 and Cj-DLP2 are monomeric in solution but form a 2:2 stoichiometric tetramer, termed Cj-DLP1/2$_{tetramer}$, when mixed (left panel). Corresponding SDS–PAGE Coomassie and SYPRO staining of Cj-DLP1/2$_{tetramer}$ with stoichiometry (right panel)

state[18]. The mitofusins or OPA1 family have tip helices that provide permanent membrane attachment. For the mitofusin homologue Fzo1p, the equivalent transmembrane region protrudes into the intermembrane space[19]. Functional tuning also occurs by domain addition or omission[20]. Fzo1p has a poorly understood but functionally important N-terminal 190 amino acid (aa) domain[21]. Similarly, additional domains are common amongst bacterial DLPs[8].

Emerging as fundamental to DLP-mediated membrane fission and fusion is a high level of inter-domain flexibility within the DLP subunit. BDLP1 nucleotide and lipid binding induces a 135° rotation from the closed to open conformation around hinge 1, and a 75° rotation from open to closed conformation around hinge 2[18] (Fig. 1a). Conformational changes around hinge 2 have been described in both classical dynamins and mitochondrial DLPs between the G-domain and bundle signalling element (BSE)[22–25]. There is also nascent evidence for conformational change around hinge 1 in eukaryotic DLPs[16,21,26–28]. The hinge regions and conformational changes described for BDLP1 are therefore emerging as conserved, albeit with modification, amongst many dynamin family members. However, the precise conformation relative to nucleotide state (or lipid binding) is not conserved amongst DLPs and instead appears tuned to specific membrane remodelling function and mechanism.

The recent partial structure of Mitofusin 1 showed the GTPase and HD1 domain to be in the BDLP1 hinge 2 open conformation[26,29]. Based on Asp189 forming an inter-domain tether, Mitofusin 1 is also predicted to fold at hinge 1 like BDLP1[26]. Together the Mitofusin 1 and BDLP1 structures hint at a molecular mechanism for membrane fusion, where opposing membranes must be recruited, tethered, and brought into close apposition before physical merging[30]. How membrane tethering is initiated has been a long-standing question. An early model was based on the anti-parallel association of the mitofusin HR2 helix[31]. More recently it has been proposed that the HR2 helix might unfold from the HD1 and HD2 domains[32]. Alternatively, the partial Mitofusin 1 structure bound to GDP·AlF4⁻ suggested tethering may occur via G-dimerisation[26]. This model is supported by atlastin membrane fusion models where G-dimerisation couples opposing membranes together[33]. In the case of Mitofusin 1 a tetramer may constitute the oligomeric state competent to tether membranes[13]. Similarly, no structural data is available for how heterotypic Mitofusin 1 and 2 complexes self-associate and orchestrate membrane fusion.

Given the mitofusins form functionally important homotypic and heterotypic oligomers between isoform pairs, and that bacterial DLPs may also require heterotypic oligomerisation for function, we searched for a bacterial DLP pair with which to probe the mechanism of heterotypic DLP-mediated membrane remodelling. Here we structurally and biochemically characterise the DLP pair from *Campylobacter jejuni*. We show how the DLPs oligomerise into a stoichiometric tetramer that comprises a highly dynamic tripartite chain. Combined with GTPase activity and lipid binding, we present a mechanism for how the tetramer tethers and bridges distantly opposing membranes. Ultimately, the tractability of bacterial systems makes them well suited for dissecting out the fundamental principles that underlie DLP-mediated membrane remodelling.

## Results

***Campylobacter jejuni* DLPs form a tetrameric complex.** The dynamin-like genes *cj0411* and *cj0412* from *Campylobacter jejuni* share 24% sequence identity when aligned from the GTPase Switch 1 (Fig. 1b, c). *cj0411* is a putative pathogenicity factor required for host colonisation[34]. The respective proteins, here

called Cj-DLP1 (84.5 kDa) and Cj-DLP2 (71.2 kDa), were cloned, expressed and purified (Fig. 1d). Both DLPs are monomeric in solution and form a stable heterotypic tetramer, termed Cj-DLP1/2$_{tetramer}$, with 2:2 stoichiometry when mixed (Fig. 1e).

**The Cj-DLP1/2$_{tetramer}$ structure reveals a 2:2 ratio tetramer.** Crystals of Cj-DLP1/2$_{tetramer}$ were obtained in the apo state and the structure solved (Table 1, Supplementary Fig. 1 and 2). As observed in solution, the asymmetric unit of Cj-DLP1/2$_{tetramer}$ comprises a heterotypic tetramer in 2:2 ratio (Fig. 2). Both Cj-DLP1 and Cj-DLP2 have a core DLP-like fold with GTPase, neck and trunk domains. The tetramer is dominated by a 2-fold symmetry where Cj-DLP2 with its symmetry mate (termed Cj-DLP2$_\alpha$ and Cj-DLP2$_\beta$) form a central back-to-back *cis* dimer flanked on each side by Cj-DLP1 (termed Cj-DLP1$_\alpha$ and Cj-DLP1$_\beta$). Crystals of Cj-DLP1/2$_{tetramer}$ bound to GDP and GDP·AlF4⁻ (Table 1, Supplementary Fig. 3) were also obtained in the same P2₁2₁2₁ space group as for the apo state. For both, the overall model is essentially identical to apo except for the coordination of GDP within the nucleotide binding pockets. Due to slightly lower resolution achieved, the GDP structure was not refined and reported. Within the GDP·AlF4⁻ crystal, no density for the AlF4⁻ moiety was observed. Based on the similarity between the GDP and GDP·AlF4⁻ electron densities, it is understood that the GDP·AlF4⁻ structure reported here is likely representative of the GDP conformation. Crystal packing rather than nucleotide state appears to dominate the conformation, as observed with the DLP atlastin in the presence of GDP[33], and DNM1-L in the apo and GMPPNP bound state[35].

**Structural analysis of Cj-DLP2 within Cj-DLP1/2$_{tetramer}$.** With respect to Cj-DLP2, the trunk and neck are in an open

### Table 1 Data collection and refinement statistics

| | Apo | GDP·AlF4⁻ | SeMet |
|---|---|---|---|
| **Data collection** | | | |
| Space group | P2₁2₁2₁ | P2₁2₁2₁ | P2₁2₁2₁ |
| Cell dimensions | | | |
| $a, b, c$ (Å) | 112.6, 226.1, 317.9 | 114.7, 228.7, 318.8 | 119.9, 226.4, 313.1 |
| $\alpha, \beta, \gamma$ (°) | 90, 90, 90 | 90, 90, 90 | 90, 90, 90 |
| Resolution (Å) | 3.7 (3.8–3.7)ᵃ | 3.9 (4.0–3.9) | 6.0 (6.6–6.0) |
| $R_{merge}$ (%) | 8.9 (147.5) | 9.4 (81.1) | 8.5 (80.5) |
| $I/\sigma I$ | 11.1 (1.2) | 14.6 (1.3) | 14.9 (2.5) |
| CC$_{1/2}$ | 99.9 (50.8) | 99.7 (63.5) | 99.8 (86.7) |
| Completeness (%) | 99.7 (99.6) | 95.5 (79.6) | 99.7 (100) |
| Redundancy | 4.9 (5.0) | 3.8 (3.9) | 6.8 (7.0) |
| **Refinement** | | | |
| Resolution (Å) | 106.2–3.7 | 59.0–3.9 | |
| No. reflections | 85,496 | 71,587 | |
| $R_{work}/R_{free}$ (%) | 25.9/28.6 | 25.9/28.8 | |
| No. atoms | | | |
| Protein | 19,375 | 19,475 | |
| Ligand/ion | 0 | 112 | |
| Water | 0 | 0 | |
| *B*-factors | | | |
| Protein | 212.5 | 232.2 | |
| Ligand/ion | | 237.1 | |
| Water | | | |
| R.m.s. deviations | | | |
| Bond lengths (Å) | 0.003 | 0.004 | |
| Bond angles (°) | 0.858 | 1.258 | |

ᵃValues in parentheses are for highest-resolution shell

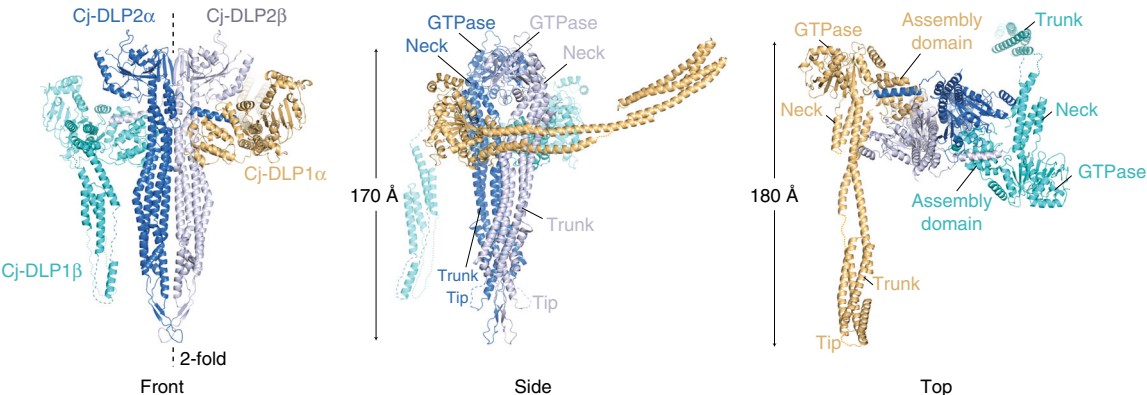

**Fig. 2** The crystal structure of Cj-DLP1/2$_{tetramer}$ in the apo state. Cj-DLP1 and Cj-DLP2 exist as a tetrameric oligomer with 2:2 stoichiometry within the asymmetric unit. Cj-DLP2 forms a central back-to-back dimer flanked on each side by Cj-DLP1 subunits. Cj-DLP1 subunits are diametrically opposed with trunk domains observed in radically different conformations. Tip region represents equivalent position of the BDLP lipid binding paddle[9]

**Fig. 3** Structural analysis of Cj-DLP2 bound to GDP·AlF$_4^-$. **a** Cartoon comparison of Cj-DLP2 monomer with the ETEC DLP LeoA. Blue represents N-terminus through to the red C-terminus. Cj-DLP2 and LeoA have similar neck and trunk architecture with RMSD = 7.0 Å when superimposed (right panel). No distinct hinge 1 region is observed at the neck-trunk interface. The Cj-DLP2 GTPase domain is in a hinge 2 closed position and requires a complex two plane twist to generate the LeoA GTPase domain conformation (open position). The lipid binding domain includes residues 530FVLF534 and is disordered in the model. **b** SEC-MALS shows that removal of the Cj-DLP2 trunk tip (amino acids 348–401 and 509–542 coloured in red) abolishes Cj-DLP2 back-to-back *cis* dimerisation within Cj-DLP1/2$_{tetramer}$, and results in Cj-DLP1/2 heterodimer instead. **c** Focus on the Cj-DLP2 back-to-back *cis* dimer showing all structural domains including the GTPase, neck and trunk contribute to the 5969 Å² dimerisation interface. BDLP1 forms a similar back-to-back dimer when self-assembled as derived from the EM fitted model[18]

conformation (Fig. 1a) and together form a linear series of 4-helix bundles. Similar architecture has been observed in the poorly understood DLP LeoA[4] despite a low shared sequence identity of <20% (Fig. 3a). However, Cj-DLP2 and LeoA differ significantly in the conformation of their GTPase domains. For Cj-DLP2,

hinge 2 exists in the closed position (Fig. 1a) so that the GTPase domain (G-domain) nucleotide-binding pocket is oriented orthogonal to the neck-trunk long axis. This arrangement is reminiscent of, and confirms, the cryo-EM modelled conformation of BDLP1 when self-assembled[18]. For LeoA, hinge 2 is in an

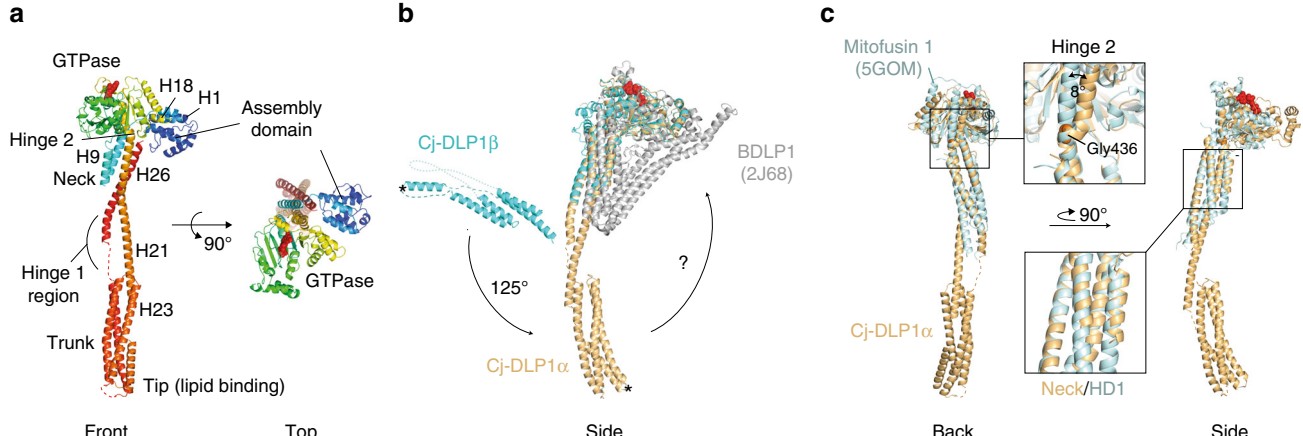

**Fig. 4** Structural analysis of Cj-DLP1 bound to GDP·AlF$_4^-$. **a** Cartoon of Cj-DLP1 monomer showing a BDLP1-like fold augmented with an N-terminal globular assembly domain (Cj-DLP1$_{AD}$). Like BDLP1, Cj-DLP1 has a distinct highly flexible hinge 1 region. Here, hinge 1 and hinge 2 are in the open conformation. **b** G-domain superposition of Cj-DLP1$_\alpha$, Cj-DLP1$_\beta$, and BDLP1[9]. Hinge 1 facilitates a 125° in-plane rotation between symmetry mates. * = equivalent positions. Freedom of movement around hinge 1 provides a mechanism for lipid binding in almost any orientation relative to the G-domain. Whether Cj-DLP1 hinge 1 folds like BDLP1 remains to be determined. **c** Cartoon superposition of Cj-DLP1$_\alpha$ and Mitofusin 1 partial structure[26]. Cj-DLP1$_{AD}$ has been removed for clarity. Neck and equivalent HD1 domains were aligned and show high similarity with helices running in phase. G-domains are offset by an 8° rotation around Gly436. Structural and conformational similarity between Cj-DLP1 and Mitofusin 1 suggests Cj-DLP1 is likely representative of full length Mitofusin 1

open position so that a substantial screw-like rotation reorients the GTPase domain with the nucleotide-binding pocket facing outwards along the neck long axis (Fig. 3a). Taken together, Cj-DLP2 and LeoA appear to represent a bacterial DLP subtype now captured in two different conformational states. For the back-to-back *cis* Cj-DLP2 dimer, the dimerisation interface extends along each symmetry mate so that the GTPase, neck and trunk domains contribute to a substantial buried surface area of 5969 Å$^2$. Removal of amino acids 348–401 and 509–542 within the trunk tip, to create Cj-DLP2$_{\Delta trunk-tip}$, is sufficient to inhibit self-assembly via the dimerisation interface so that only Cj-DLP1/2 heterodimers assemble (Fig. 3b). This result, combined with the observation that the BDLP1 polymer utilises a similar back-to-back dimer (Fig. 3c)[18], indicates that the crystallographic configuration of Cj-DLP2 homodimer is conserved and important for function.

**Structural analysis of Cj-DLP1 within Cj-DLP1/2$_{tetramer}$.** Cj-DLP1 has a canonical BDLP1-like fold augmented with an N-terminal 119 amino acid globular domain that nestles between the neck and G-domains (Fig. 4a). As described later, this domain is essential for oligomerisation and is termed the assembly domain, or Cj-DLP1$_{AD}$. Cj-DLP1 differs from the Cj-DLP2/LeoA subtype as its neck and trunk 4-helix bundles do not overlap and instead are linked by a distinct hinge 1 region within helices 21 and 26. Hinge 1 acts as a fulcrum with the trunks of Cj-DLP1$_\alpha$ and Cj-DLP1$_\beta$ assuming significantly different conformations relative to each other. Cj-DLP1$_\alpha$ forms an open linear conformation (Fig. 1a), whereas the Cj-DLP1$_\beta$ trunk is rotated in plane ~125° towards the neck C-terminus (Fig. 4b). While the trunk orientation is likely determined by crystal geometry here, the conformational variability reflects the flexibility of the hinge 1 neck-trunk interface. Importantly, such flexibility suggests that membrane will be sensed and bound by the lipid-binding domain at the trunk tip in almost any orientation relative to the neck and GTPase domain. Compared with Cj-DLP2, the G-domain of both Cj-DLP1$_\alpha$ and Cj-DLP1$_\beta$ is rotated in-plane almost 90° around hinge 2. This open conformation (Fig. 1a) means that the nucleotide-binding pocket is oriented facing outwards along the neck long axis, and is similar to that observed for BDLP1 in the apo- or GDP-bound state[9], and Mitofusin1 partial structures[26,29].

Importantly, full-length Cj-DLP1$_\alpha$ represents the conformation predicted for Mitofusin 1 to tether opposing membranes upon G-dimer formation (Fig. 4c). These results, combined with the observation that the yeast mitofusin homologue Fzo1p[36] has an N-terminal heptad repeat domain (HRN) of similar mass to Cj-DLP1$_{AD}$, suggests Cj-DLP1$_\alpha$ is a close respresentation of the mitofusin/Fzo1p class of DLPs.

**Cj-DLP1/2$_{tetramer}$ oligomerisation mechanism.** The structure of Cj-DLP1/2$_{tetramer}$ provides a mechanism for oligomerisation. The N-terminal 16 amino acids of Cj-DLP2 form helix 1 (Cj-DLP2$_{H1}$), which inserts into a groove within Cj-DLP1$_{AD}$ (Fig. 5a). Here, Cj-DLP2$_{H1}$ is partially clamped by Cj-DLP1 G-domain helix 18 and terminates in close proximity to Cj-DLP1 hinge 2, meaning it is ideally positioned to modulate Cj-DLP1 conformation upon binding. Truncation of either Cj-DLP2$_{H1}$ or Cj-DLP1$_{AD}$ inhibits tetramer formation and results in monomer (Fig. 5b). This shows Cj-DLP2$_{H1}$ and Cj-DLP1$_{AD}$ interaction to be essential for Cj-DLP1 and Cj-DLP2 heterodimerisation and subsequent tetramer formation. Within the Cj-DLP1/2$_{tetramer}$ asymmetric unit, the other potential oligomerization interface between Cj-DLP1 and Cj-DLP2 is the contact between Cj-DLP1$_{AD}$ and Cj-DLP2 at its neck and trunk interface (Supplementary Fig. 4a). However, given the limited contact area (494 Å$^2$), the geometry of the P2$_1$2$_1$2$_1$ space group with diametric arrangement of Cj-DLP1 symmetry mates, and the absence of heterodimer in the Cj-DLP2$_{H1}$ truncation, this interface is considered a crystal packing contact only. Connection between the Cj-DLP2$_{H1}$/Cj-DLP1$_{AD}$ complex and the bulk of Cj-DLP2 is therefore via a 9 amino acid random coil linker only (termed Cj-DLP2$_{linker}$) (Fig. 5c). Such architecture suggests that Cj-DLP2$_{linker}$ represents a flexible tether in which Cj-DLP1$_\alpha$ and Cj-DLP1$_\beta$ will be free to explore, in a mode akin to restricted Brownian motion[37], a spectrum of positions and orientations relative to the central Cj-DLP2 dimer. Of significance is whether Cj-DLP2$_{linker}$ extends sufficiently to allow Cj-DLP1 and Cj-DLP2 G-domains to heterodimerise across the nucleotide-binding pocket interface. G-dimerisation is widely conserved in other DLPs and protein classes such as septins and represents a fundamental mechanism for increasing catalysis rate[38]. Molecular dynamics (MD) simulations modelling the movement of Cj-DLP1/2$_{tetramer}$ in solution show that Cj-DLP2$_{linker}$ is indeed

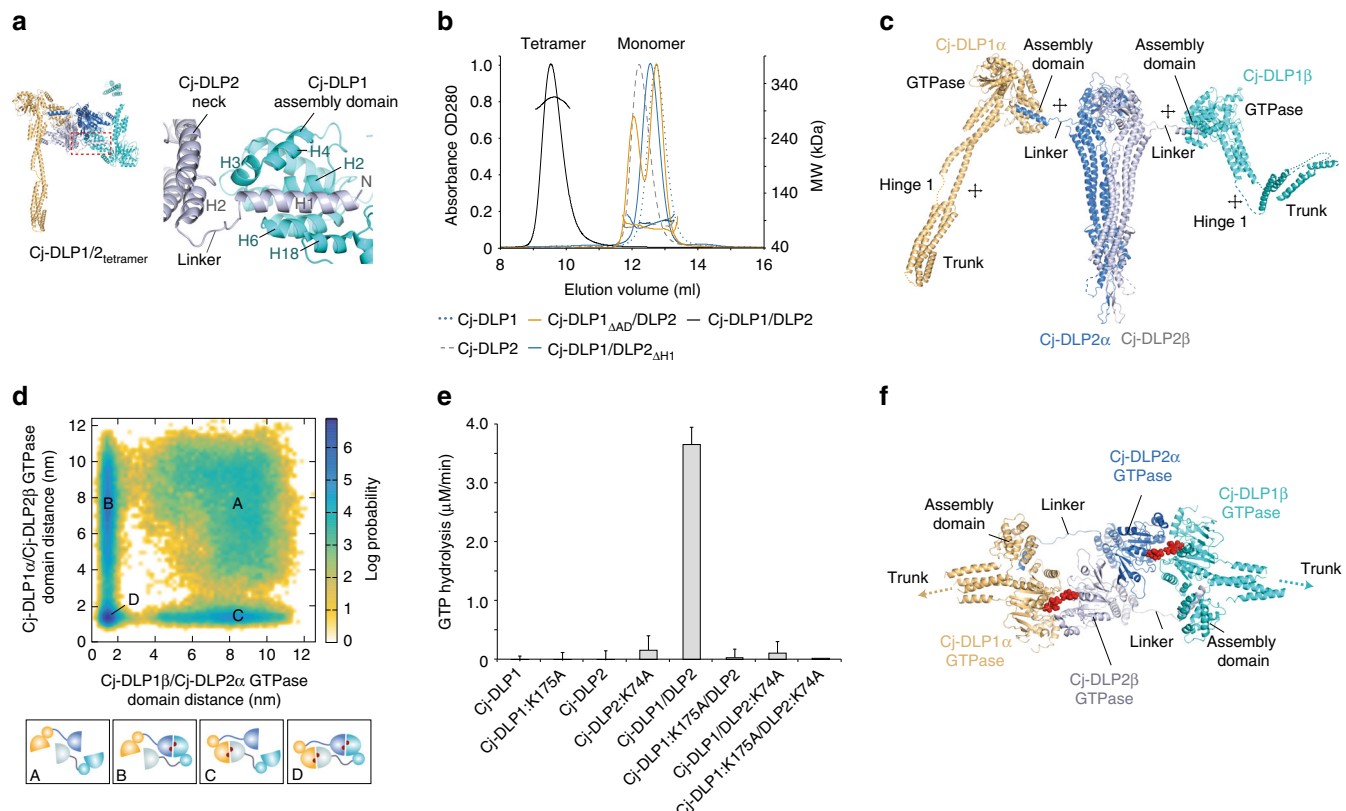

**Fig. 5** Cj-DLP1/2$_{tetramer}$ oligomerisation and catalysis. **a** Cartoon showing Cj-DLP1 and Cj-DLP2 heterodimerisation mechanism. The N-terminal Helix 1 of Cj-DLP2 inserts into a groove within Cj-DLP1$_{AD}$. Right panel shows zoom of left panel box. **b** SEC-MALS shows that truncation of Cj-DLP1$_{AD}$ or Cj-DLP2$_{H1}$ abolishes tetramer formation. **c**, Schematic model of Cj-DLP1/2$_{tetramer}$ when in a solution state, rather than crystalline. The position of each Cj-DLP1 subunit relative to Cj-DLP2 has been chosen arbitrarily to highlight the 9 amino acid random coil linker. Crossed arrows indicate regions of high flexibility. **d** Histogram showing a structure-based molecular dynamics (MD) simulation of the Cj-DLP1/2$_{tetramer}$ in **c**. The Cj-DLP2$_{linker}$ is of sufficient length for GTPase heterodimerisation to occur. The two G-domain pairs transition between apo state in A, a mix of apo state and G-dimer in B and C, or G-dimer in D. **e** GTPase assays show that Cj-DLP1/2$_{tetramer}$ exhibits significant assembly stimulated turnover which is a known catalytic consequence of G-dimerisation. 1 or 2 μM protein concentration was used for tetrameric or monomeric species, respectively. Data represent mean ± standard deviation from 2 independent experiments. **f** Model of Cj-DLP1/2$_{tetramer}$ G-dimerisation based on data presented in **d** and **e**. Note that the geometry of the linker allows no other combination of G-dimerisation to occur within Cj-DLP1/2$_{tetramer}$ in the observed conformations

sufficient to allow G-dimer formation between Cj-DLP1 and Cj-DLP2 (Fig. 5d, Supplementary Fig. 4b and Supplementary Movie 1). This in silico model is supported by in vitro GTPase activity assays. While individual Cj-DLP1 and Cj-DLP2 proteins show negligible GTP hydrolysis, Cj-DLP1/2$_{tetramer}$ at 1 μM concentration shows significant assembly-stimulated activity with a maximal observed hydrolysis rate of 3.6 μM/min (Fig. 5e). Mutations generated in both P-loops (Cj-DLP1/K175A and Cj-DLP2/K74A) or in individual P-loops within Cj-DLP1/2$_{tetramer}$, show negligible GTP turnover (Fig. 5e and Supplementary Fig. 4c). This means that for Cj-DLP1/2$_{tetramer}$, the GTP hydrolysis mechanism depends on heterotypic G-dimerisation between Cj-DLP1 and Cj-DLP2. Moreover, the presence of assembly stimulated GTP turnover specifically supports a model where catalysis is triggered by G-dimer formation across the nucleotide-binding pocket interface[22] (Fig. 5f). G-dimerisation between, rather than within, discrete Cj-DLP1/2$_{tetramer}$ complexes should occur increasingly at higher protein concentrations when the effect of Cj-DLP2$_{linker}$ to maintain high local GTPase domain concentration becomes less relevant.

**Cj-DLP1/2$_{tetramer}$ binds lipid in the absence of nucleotide.** A direct role for membrane binding and remodelling was tested for in vitro. At 0.25–1 μM protein concentrations, spin assays show

that Cj-DLP1 binds weakly to *E. coli* liposomes, while Cj-DLP2 exhibits negligible binding (Fig. 6a and Supplementary Fig. 5a). Increasing concentration to 10–20 μM shows equivalent lipid binding for both Cj-DLP1 and Cj-DLP2 although the binding efficiency remains relatively low (Supplementary Fig. 5b). For Cj-DLP2, the FVLF530-534EEEE mutation in the trunk tip inhibits lipid binding (Supplementary Fig. 5b) and suggests this region is the equivalent to the BDLP1 paddle. In contrast to individual Cj-DLP1 and Cj-DLP2 samples, Cj-DLP1/2$_{tetramer}$ shows efficient liposome binding at low concentration (0.5 μM). The addition of nucleotide including GTP, GMPPCP, and GDP has no obvious effect (Fig. 6a). Deletion of Cj-DLP1 amino acids 470–695 from Cj-DLP1/2$_{tetramer}$ to generate Cj-DLP1$_{Δtrunk}$/2$_{tetramer}$ (Fig. 6b) removes the predicted lipid binding region in the trunk tip and abrogates liposome binding (Fig. 6a and Supplementary Fig. 5b). At low concentration (0.5 μM), Cj-DLP1/2$_{tetramer}$ recruitment to the membrane is therefore mediated by Cj-DLP1.

**Cj-DLP1/2$_{tetramer}$ remodels and tethers membrane.** At high sample concentration (10–20 μM) and in the presence of liposomes, Cj-DLP1, Cj-DLP2 and Cj-DLP1/2$_{tetramer}$ form poorly ordered protein–lipid tubes ~40–50 nm in diameter that appear broadly similar in architecture (Fig. 6c). Liposome tubulation may be indicative of helical filament formation or the effect of protein

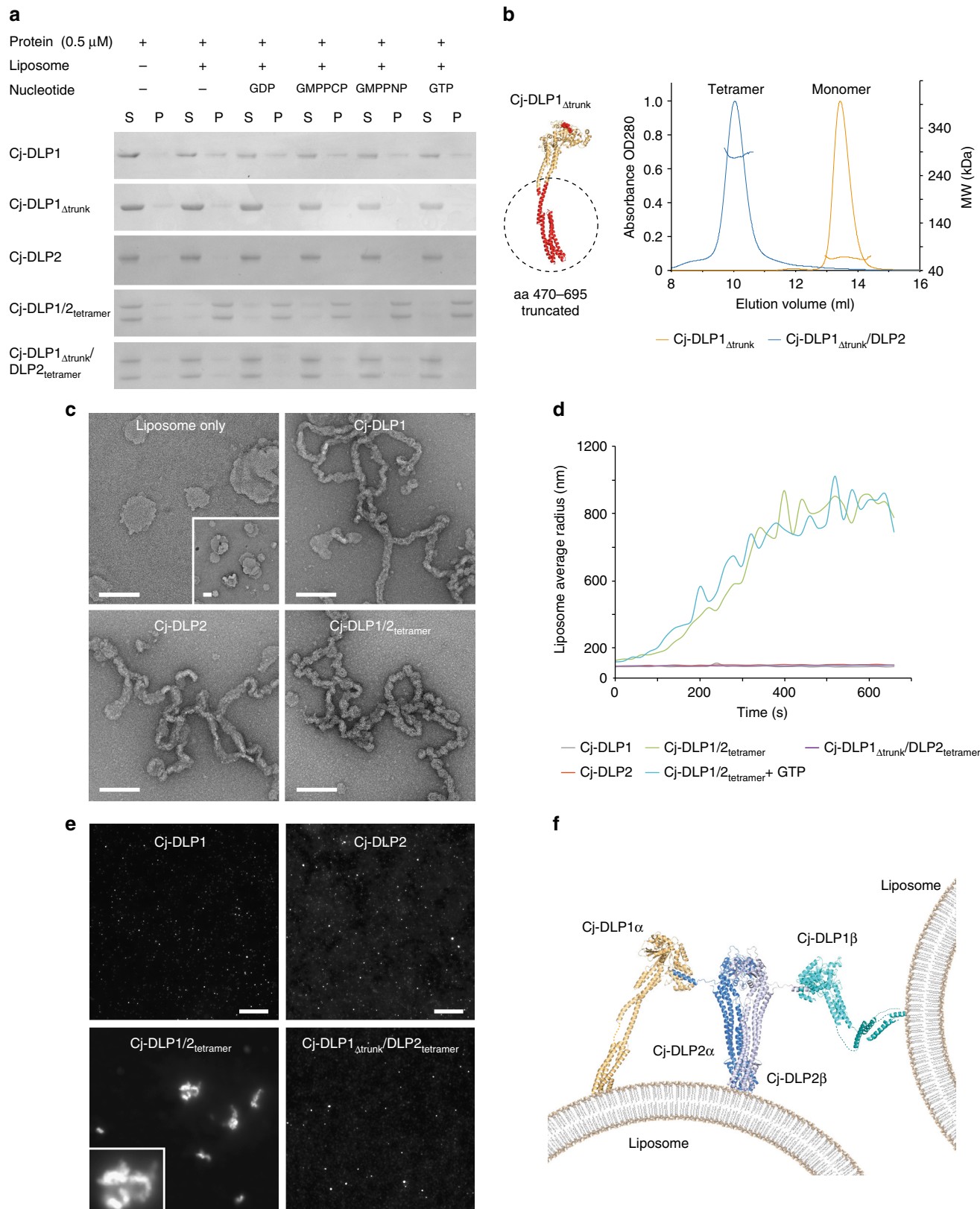

crowding at high concentration. The addition of nucleotide had no obvious effect on the tube morphology. Working now at low concentration (0.5 μM), dynamic light scattering (DLS) was used to detect membrane tethering or fusion in vitro. The mixing of ~70 nm radius liposomes with Cj-DLP1, Cj-DLP2, or Cj-

DLP1$_{\Delta trunk}$/DLP2$_{tetramer}$ had no effect on average liposome size. In contrast, the mixing of Cj-DLP1/2$_{tetramer}$, in the presence or absence of GTP, increased average liposome diameter at least 10-fold (Fig. 6d), which is indicative of liposome tethering and possibly fusion. This result was confirmed using fluorescent light

**Fig. 6** Cj-DLP1/2$_{tetramer}$ binds, tubulates and tethers liposomes *in trans*. **a** Spin assays show that only Cj-DLP1/2$_{tetramer}$ binds lipid efficiently at low (0.5 μM) concentration. The addition of nucleotide has no obvious effect. Removal of the Cj-DLP1 trunks to create Cj-DLP1$_{\Delta trunk}$/DLP2$_{tetramer}$ inhibits lipid binding. S = supernatant, P = pellet. **b** SEC-MALS shows that mixing of Cj-DLP1$_{\Delta trunk}$ and Cj-DLP2 maintains a Cj-DLP1/2$_{tetramer}$-like assembly. **c** Cj-DLP1, Cj-DLP2 and Cj-DLP1/2$_{tetramer}$ bind and tubulate lipid at high protein concentration (≥10 μM). Tube morphology appears overall similar across all samples and irrespective of nucleotide addition. Scale bar = 200 nm. **d** Dynamic light scattering plot shows significant Cj-DLP1/2$_{tetramer}$-mediated liposome tethering or fusion. Tethering is not observed with Cj-DLP1$_{\Delta trunk}$/DLP2$_{tetramer}$. **e** Fluorescent microscopy shows that only Cj-DLP1/2$_{tetramer}$ is competent to induce liposome tethering or fusion. Scale bar = 10 μM. Inset panel = 4 μM. **f** Cartoon model showing the mechanism of nucleotide free Cj-DLP1/2$_{tetramer}$-mediated liposome tethering at low concentration (0.5 μM)

microscopy where the same samples were mixed with liposomes incorporating 4% rhodamine-PE. While Cj-DLP1, Cj-DLP2, or Cj-DLP1$_{\Delta trunk}$/DLP2$_{tetramer}$ samples yielded monodisperse proteoliposomes similar to native liposomes, those containing Cj-DLP1/2$_{tetramer}$ formed easily visible membrane clusters and aggregates (Fig. 6e). The equivalent samples were visualised by negative stain EM to similar effect (Supplementary Fig. 5c). Membrane aggregation was sufficiently efficient that sample thickness hindered visualisation by cryo-EM. Given such extensive membrane aggregation, it was unclear why membrane fusion was not subsequently readily detected by FRET based liposome mixing assays[39]. Overall, these data are indicative of liposome tethering at low concentration (0.5 μM) (Fig. 6f) and suggests Cj-DLP1/2$_{tetramer}$ drives membrane fusion rather than fission reactions.

## Discussion

The structure of Cj-DLP1/2$_{tetramer}$ provides a mechanism for heterotypic oligomerisation by a DLP pair triggered by the insertion of Cj-DLP2$_{H1}$ into a groove within Cj-DLP1$_{AD}$. Oligomerisation activates both Cj-DLP1 and Cj-DLP2 triggering significant assembly stimulated GTPase activity likely via G-dimerisation. While Cj-DLP1$_{AD}$ is specific to *Campylobacter*, the equivalent heterotypic coupling of two DLPs has been achieved in other bacterial species such as *B. subtilis* DynA by the head-to-tail fusion of a DLP gene pair[5]. This would have the effect of coupling a DLP pair by a short linker similar to the Cj-DLP1/2 heterodimer. DynA also oligomerises into a dimer capable of nucleotide independent tethering of liposomes[5], which suggests that Cj-DLP1/2$_{tetramer}$ is structurally representative of this class of fused DLPs, which also includes *Staphylococcus* strains. Further studies in other bacterial systems are required to understand how extensively conserved heterodimeric coupling is. Many bacterial DLPs have gene modifications that include additional non-canonical DLP structural domains[8] that might function similarly to Cj-DLP1$_{AD}$. For example, the DLP gene neighbouring BDLP1 (uniprot B2IZD2) incorporates a DnaJ-like domain at its C-terminus that may act not only as a DnaK binding domain as its sequence predicts[40], but also as an assembly domain to dimerise with BDLP1. The observation that Cj-DLP2 and Cj-DLP1 are inactive unless both are present may now explain why the structurally similar LeoA is inactive when monomeric and suggests that LeoA will indeed bind with LeoB and LeoC for activation and function[4]. Similarly, *B. subtilis* DynA requires both DynA$_{D1}$ and DynA$_{D2}$ GTPases to be functional for activation[5].

In the absence of membrane stress, *B. subtilis* DynA localises uniformly at the inner membrane surface. In the presence of membrane pore forming agents such as nisin-like antibiotics or bacteriophage, DynA reorganises into punctate foci at the inner membrane where it is proposed to function as a fusogen in membrane maintenance and repair[6]. Recent striking images of phage endolysin-induced holes in the *B. subtilis* inner membrane[41] confirm a critical cellular requirement for a fusogenic membrane repair machine capable of tethering and sealing

distantly opposing membranes. Cj-DLP1/2$_{tetramer}$ has a cytosolic location within the *Campylobacter* cell (Supplementary Fig. 6) and a general membrane repair function is speculated as for *B. subtilis* DynA. The presence of an HcpA-like β-lactamase[42] within the *cj-dlp1/2* operon (Fig. 1b) suggests that β-lactam antibiotics as a source of cell envelope stress could be a trigger for membrane localisation. Alternative phenotypes are possible although a key role in cell division cytokinesis equivalent to *Streptomyces* DynA and DynB[3] may tentatively be excluded. No obvious cell division defects were observed in a chromosomal knockout of *cj-dlp1* and *cj-dlp2* (Supplementary Fig. 6).

Our results provide a molecular model for how a bacterial heterotypic dynamin-like pair tethers membrane bilayers (Supplementary Fig. 7). In bacteria, which generally lack cytoplasmic organelles, opposing membranes are depicted as a pore within a single membrane as for DynA at the inner membrane[6]. The recruitment and binding of Cj-DLP1/2$_{tetramer}$ to membrane is nucleotide independent and is promoted by Cj-DLP1, which binds with higher avidity than Cj-DLP2. At low concentration, our data supports a model where Cj-DLP2 acts as an adaptor to couple two Cj-DLP1 subunits together and to markedly extend their membrane binding reach. This architecture combined with the inherent flexibility of Cj-DLP1/2$_{tetramer}$ provides a mechanism for distantly opposing membranes to be tethered *in trans* with a theoretical range of 40–45 nm when fully extended. The role of nucleotide for Cj-DLP1/2$_{tetramer}$ is unclear given membrane tethering and tubulation occurs in the absence of nucleotide, and its addition in vitro has no detectable effect under the conditions screened. However, the presence of assembly-stimulated nucleotide turnover, which requires both Cj-DLP1 and Cj-DLP2 GTPase domains to be active, strongly suggests nucleotide binding triggers hetero G-dimer formation. Localised concentration of Cj-DLP1/2$_{tetramer}$ on the membrane induces polymerisation and consequently membrane constriction akin to liposome tubulation. As local concentration increases, the degree to which G-dimerisation between neighbouring Cj-DLP1/2$_{tetramer}$ complexes occurs remains to be determined. Given the close structural homology of Cj-DLP1 with human Mitofusin 1 and the propensity of Cj-DLP1/2$_{tetramer}$ to tether membranes, it is likely that ultimately Cj-DLP1/2$_{tetramer}$ will mediate membrane fusion rather than fission reactions. A membrane fusion phenotype is also consistent with that proposed for *B. subtilis* DynA[43]. Nucleotide hydrolysis may be directly coupled to active fusogenic conformational changes within Cj-DLP1/2$_{tetramer}$ such as the arcing together of Cj-DLP1 and Cj-DLP2 neck and trunks as observed in the BDLP1 polymer[18] and for Mitofusin 1[44], or their folding like a molecular staple[9]. These conformational changes may physically merge and fuse opposing membranes as suggested for Mitofusin 1[26]. Alternatively, the spontaneous formation of Cj-DLP1/2$_{tetramer}$ protein-lipid tubes in vitro at high concentration suggests that membrane fusion may be driven passively as a consequence of induced membrane curvature[18]. In this case nucleotide hydrolysis may represent a mechanism to release Cj-DLP1/2$_{tetramer}$ from the membrane, as for BDLP1. The structure of Cj-DLP1/2$_{tetramer}$ also provides a natural mechanism for the

bridging of opposing double membranes. In bacteria, double membrane systems might include inner membrane bulging induced by phage endolysin[41], cell envelope vesiculation[8] or the abscission of the cytokinetic septum during cell division. In structurally related eukaryotic DLP systems such as Fzo1p and the mitofusins, full length oligomerised structures are required to determine whether the Cj-DLP2 homodimer arrangement is conserved across evolutionary domains.

Cj-DLP1 is a BDLP1-like subtype with a distinct and highly flexible hinge 1 region between neck and trunk 4-helix bundles. The conformation of Cj-DLP1, with hinge 1 and hinge 2 open, is different to those previously observed for BDLP1. The structure is important as it represents the predicted conformation required for DLPs to bridge opposing membranes through G-dimerisation. Given the similarity of Cj-DLP1 to Mitofusin 1 partial structures, it therefore supports a mitochondrial membrane tethering model based on G-dimerisation[26] rather than the anti-parallel association[31] and the unfolding[32] of the HR2 helix. The structure is also reminiscent of the conformation of Dynamin 1 when polymerised and bound to GMPPCP, which has so far has only been described from structures fitted as rigid bodies into cryo-EM filament reconstructions[45].

Cj-DLP2 has closer structural similarity to the ETEC DLP LeoA than Cj-DLP1 and BDLP1. Distinguishing features include overlapping neck and trunk domains with a reduced hinge 1 region. While this architecture does not preclude hinge 1 closing, the overlapping neck and trunk suggests the potential for less flexibility compared with Cj-DLP1, BDLP1 and Dynamin 1. The incorporation of Cj-DLP2 into a back-to-back dimer means any hinge 1 closure would likely require dimer dissociation or significant conformational rearrangement. MD simulations also suggest that heterotypic G-dimerisation may only occur when hinge 2 is in, or near, the closed position due to the geometric constraints of Cj-DLP2$_{linker}$ (Fig. 5f). However, this does not limit the Cj-DLP2 GTPase domain from forming a LeoA-like open conformation during periods of the nucleotide hydrolysis cycle when not G-dimerised. Whether Cj-DLP2 structure is representative of Mitofusin 2, with neck and trunk arrangements closer to LeoA, remains to be determined. Minimally, the Cj-DLP2 structure with hinge 1 open and hinge 2 closed reflects the conformation of full-length Mitofusin 1 when bound to GDP.BeF$_3$[44]. A similar conformation has previously been shown for polymerised BDLP1 when bound to GMPPNP[18]. For both Cj-DLP2 and BDLP1 a back-to-back dimer is observed when oligomerised (Fig. 3c), which raises the question as to whether either Mitofusin 1 or Mitofusin 2 will utilise a similar dimerisation interface. Cj-DLP2 exists as a monomer in solution rather than as the back-to-back dimer. Such Cj-DLP2 self-assembly autoinhibition may be mediated by Cj-DLP2$_{H1}$, which in the absence of Cj-DLP1$_{AD}$ may bind *in cis* across part of the dimerisation interface and sterically hinder self-assembly. However, as truncation of Cj-DLP2$_{H1}$ yields a monomeric species and does not form dimer (Fig. 5b), it may be that a Cj-DLP2 conformational change, such as hinge 1 closure, ultimately mediates self-assembly autoinhibition.

Bacterial DLPs usually exist as at least a genetic pair within operons, and our data suggests that heterotypic pairing in bacteria is fundamental for function. Our results show a novel mechanism for oligomerisation within the dynamin family, one that facilitates the long-range recruitment and tethering of membranes. They provide a molecular context for how other bacterial DLP pairs such as *B. subtilis* DynA may operate. In addition, the structure of Cj-DLP1 is important as it shows high structural homology to Mitofusin 1, for which there is currently no full-length structure. The Cj-DLP2 structure is similar to ETEC LeoA but in an alternative conformation. The Cj-DLP2 homodimer arrangement is equivalent to polymerised BDLP1 dimers[18]. BDLP1 also has

close structural homology to Mitofusin 1. Cj-DLP1 and Cj-DLP2 therefore unify multiple bacterial and eukaryotic DLP systems, and represent an important snapshot into the type of conformations and oligomerisation mechanisms that will likely be fundamental for DLP-mediated membrane fusion.

## Methods

**Cloning, protein expression and purification.** The *cj-dlp1* gene (*cj0411*) from *Campylobacter jejuni* was cloned into pHis17 vector (pET derivative), encoding the full-length protein with a C-terminal hexahistidine tag. The *cj-dlp2* gene (*cj0412*) from *C. jejuni* was cloned into pOPTM vector (pET derivative), which encodes an N-terminal MBP fusion with a TEV cleavage site in the linker and a C-terminal hexahistidine tag. Overexpression of *cj-dlp1* and *cj-dlp2* was achieved in *E. coli* BL21 (DE3) cells induced with 1 mM IPTG at 18 °C overnight. Cells were harvested by centrifugation, flash frozen in liquid nitrogen and stored at −80 °C. A list of all primers used in this study is shown in Supplementary Table 1.

For purification of Cj-DLP1, cells were lysed by sonication in 50 mM Tris pH 8.0 and 1 M NaCl. The cell lysate was centrifuged in a Ti45 rotor (Beckman Coulter) at 98,000 × *g* at 4 °C for 45 min. The supernatant was loaded onto 2 × 5 ml HisTrap columns (GE healthcare) at 4 °C. The column was washed and eluted with the addition of 300 mM imidazole pH 8.0 to the wash buffer. 5 mM DTT was added immediately into the elution peak. The protein was gel filtrated using a HiPrep 26/60 Sephacryl S300 column (GE healthcare) in buffer TEN 7.5 (20 mM Tris pH 7.5, 1 mM EDTA, 1 mM sodium azide) containing 1 mM DTT. Fractions were concentrated to ~10 mg/ml, flash frozen in liquid nitrogen and stored at −80 °C.

For purification of Cj-DLP2, cells were lysed in 50 mM Tris pH 9.0, 1 M NaCl, 2 mM DTT and 2 mM EDTA. The lysate was centrifuged as described above and the supernatant loaded onto a self-packed column with ~10 ml of amylose resin (New England Biolabs). The column was washed with 50 mM Tris pH 9.0, 0.5 M NaCl, 1 mM DTT, 1 mM EDTA and 20% glycerol. The sample was eluted by the addition of 10 mM maltose to the wash buffer. The MBP was cleaved by TEV protease and the products separated by gel filtration using a HiPrep 26/60 Sephacryl S300 column in buffer TEN 7.5 containing 1 mM DTT. Fractions were concentrated to ~10 mg/ml, flash frozen in liquid nitrogen and stored at −80 °C.

For purification of Cj-DLP1/2$_{tetramer}$, purified Cj-DLP1 and Cj-DLP2 were mixed at equal molar ratio and incubated at 4 °C for 30 min. The resulting product was gel filtrated with a HiPrep 16/60 Sephacryl S300 column in buffer TEN 7.5 containing 1 mM DTT. Fractions were concentrated to ~10 mg/ml, flash frozen in liquid nitrogen and stored at −80 °C.

Selenomethionine labelled proteins were overexpressed in B834 (DE3) cells (methionine auxotrophic) in SelenoMethionine Medium (Molecular Dimensions) and were purified as for native.

Using the wild type Cj-DLP1 or Cj-DLP2 expression plasmid as the template, point mutations and truncations of *cj-dlp1* or *cj-dlp2* were carried out using one-step isothermal DNA assembly (Gibson assembly)[46]. All mutants and truncations were expressed and purified as for native.

**Protein crystallisation and cryoprotection.** In total 10 mg/ml Cj-DLP1/2$_{tetramer}$ was incubated with 3 mM MgCl$_2$, 2 mM GDP, 2 mM AlCl$_3$ and 20 mM NaF (abbreviated as GDP·AlF$_4^-$) for 30 min at room temperature. Equal volumes of the protein and the crystallisation solution (0.8 M succinate pH 7.0) were mixed to set up sitting-drop vapour diffusion at 20 °C. Initial crystals were harvested into 0.5 ml of reservoir solution containing GDP·AlF$_4^-$, followed by vortexing with MicroSeed Beads (Molecular Dimensions). This seed crystal stock was aliquoted, flash frozen in liquid nitrogen and stored at −80 °C. For obtaining Cj-DLP1/2$_{tetramer}$ GDP·AlF$_4^-$ crystals, the seed stock was diluted 1:100 into a pre-mixed protein-reservoir drop containing 0.7–1.0 M succinate pH 7.0 and GDP·AlF$_4^-$. Crystals appeared overnight and were harvested after one week at 20 °C. For obtaining Cj-DLP1/2$_{tetramer}$ apo crystals, the Cj-DLP1/2$_{tetramer}$ GDP·AlF$_4^-$ seeds were diluted 1:50 into a pre-mixed protein-reservoir drop containing 0.7–1.0 M succinate pH 7.0 only. Crystals obtained were then used to make a new apo seed stock with the same reservoir solution. The new apo seeds were used to grow Cj-DLP1/2 apo crystals. Iterative rounds of seed preparation from newly grown crystals did not improve the diffracting power of either the apo or GDP·AlF$_4^-$ crystals. Selenomethionine crystals were similarly prepared except using selenomethionine labelled sample. Successful cryogenic protection required a slow iterative addition of glucose to the crystallisation drop, up to 60% final concentration. The succinate concentration ± GDP·AlF$_4^-$ was maintained throughout. Crystals were harvested and flash frozen in liquid nitrogen.

**Data collection and structure determination.** Diffraction data were collected at 100 K at Diamond Light Source, UK. A SAD dataset was collected from a single crystal. One line scan dataset was collected from a single Cj-DLP1/2 apo crystal. XDS[47] and Aimless[48] were used for data integration and scaling. For GDP·AlF$_4^-$ incorporated crystals, line scan was used to collect multiple datasets of small wedges at various positions within each crystal. XDS was used to process and integrate each dataset. Blend[49] was used to analyse the degree of isomorphism

among different datasets. Utilising Aimless, nine wedge datasets from two different crystals were merged and scaled together.

Using the SAD dataset, an initial 6 Å resolution electron density map was generated by Phenix AutoSol[50]. Initially, 26 selenomethionine sites from 28 were identified. Using Phenix, non-crystallographic symmetry averaging with phase extension against higher resolution native datasets followed by density modification generated a map of sufficient quality for initial poly-alanine model building. Combined MR-SAD with the poly-alanine model allowed the identification of 27 selenomethionine sites from the SAD dataset. A new round of phase extension and density modification resulted in an improved map with good side chain detail. Iterative rounds of model building and refinement were then carried out in Coot[51], Phenix, and Refmac5[52]. Phenix_rosetta refinement[53] was used to improve model geometry. The density for Cj-DLP1$_\alpha$ hinge 1 region and trunk does not show side chain detail but is of sufficient quality to allow almost continual poly-alanine main chain to be built with confidence (residues 466–703 with chain breaks at 506–514 and 626–629). Selenomethionine residue 654 allowed unambiguous location and orientation of the trunk 4-helix bundle. To aid model building in this region a homology model based on the crystal structure of the BDLP1 (PDB ID 2J69) trunk domain was used as a template. The density for Cj-DLP1$_\beta$ hinge 1 region and trunk is of lower quality but sufficient for unambiguous location of the trunk 4-helix bundle motif. Here, the Cj-DLP1$_\alpha$ trunk model was rigid body fitted and regions with no supporting electron density removed. 90.4% of all residues are positioned within the favoured region of the Ramachandran plot with 0.74% outliers. For the GDP-AlF$_4^-$ dataset, the GDP coordinates were manually fitted into the $F_o–F_c$ density in Coot and refined. No clear density was observed for the AlF$_4^-$ moiety. For the GDP·AlF$_4^-$ model, 90.6% of all residues are positioned within the favoured region of the Ramachandran plot with 0.94% outliers.

**SEC-MALS.** To determine the mass of protein samples, an Agilent 1260 equipped with miniDAWN TREOS light scattering detector and an Optilab T-rEX refractive index detector (Wyatt Technologies) was used. 100 µl protein samples at 1–2 mg/ml were injected onto a Superdex 200 10/300 GL column (GE Healthcare) equilibrated in TEN 7.5, 1 mM DTT. Data were analysed using the ASTRA software (Wyatt Technologies).

**Liposome preparation.** *E. coli* whole cell lipid (Avanti Polar Lipids) was used to make liposomes. For making fluorescent liposomes, rhodamine-PE (Thermo Scientific) was mixed with *E. coli* whole cell lipid in a molar ratio of 4:96 in chloroform. A nitrogen stream was used to evaporate the chloroform and the lipid film then dried under vacuum for 3 h. The lipid film was subsequently re-suspended in 20 mM HEPES pH 7.2, 100 mM NaCl and 1 mM DTT (termed reaction buffer). The resulting two-phase mixture was sonicated and then extruded using a polycarbonate membrane with pore size 0.2 µm (Mini-Extruder, Avanti Polar Lipids).

**Spin assays.** Cj-DLP1, Cj-DLP2 or Cj-DLP1/2$_{tetramer}$ samples at the desired concentration were incubated ± 0.3 mg/ml *E. coli* whole cell liposomes in reaction buffer at room temperature for 30 min. 3 mM MgCl$_2$ and 2 mM nucleotide (GDP, GTP, GMPPCP, GMPPNP) were added accordingly. Reactions were spun in a TLA 100 rotor (Beckman Coulter) at 20 °C for 20 min at 100,000 × g using an Optima TLX100 ultracentrifuge (Beckman coulter). Supernatants and pellets were analysed by SDS–PAGE, followed by silver staining (SilverQuest™ Silver Staining Kit, Invitrogen). All experimental conditions were repeated at least twice.

**EM analysis of Cj-DLP1/2 mixed with liposomes.** 0.5–20 µM Cj-DLP1, Cj-DLP2 or Cj-DLP1/2$_{tetramer}$ samples were mixed with 0.3–1 mg/ml *E. coli* whole cell liposomes at room temperature for up to 4 h. The samples were applied to glow-discharged, 400-mesh carbon coated copper grids and stained with 2% uranyl acetate. Images were recorded on a CM200 (Philips) and T12 Spirit equipped with 4 K CCD camera.

**Liposome tethering assay.** The DLS tethering assay was carried out as described for atlastin[54]. Zetasizer (Malvern) was used with default settings to determine liposome size in solution. 1 µM Cj-DLP1, Cj-DLP2 or Cj-DLP1/2$_{tetramer}$ samples were mixed with 40 µl 0.3 mg/ml *E. coli* whole cell liposomes at room temperature. 3 mM MgCl$_2$ and 2 mM GTP were added as required. The same samples were also visualised by fluorescent microscopy using a Leica DMi8 with ×40 objective and a DFC365 FX camera. Liposomes had 4% rhodamine-PE incorporated. After 30 min incubation at room temperature, the samples were spotted onto a glass coverslip and imaged. All experimental conditions were repeated at least twice.

**GTPase assay.** For analysing GTPase activity, 2 µM Cj-DLP1 and Cj-DLP2 samples, or 1 µM Cj-DLP1/2 tetramer complexes were used. Reactions at 37 °C were carried out in the reaction buffer and initiated via the addition of 1 mM GTP and 2.5 mM MgCl$_2$, and quenched with the addition of 0.1 M EDTA pH 8.0. Free phosphate concentration was determined using a malachite green based kit (PiColorLock™ Gold Phosphate Detection System, Innova Biosciences).

**Campylobacter growth and gene knockout.** Wild type *C. jejuni* 81–176 were grown on MH agar under micro-aerobic conditions using CampyGen sachets (Oxoid) at 37 °C, as described previously[55]. For knockout of *cj-dlp1* (*cj0411*) and *cj-dlp2* (*cj0412*) genes from the *C. jejuni* chromosome, the whole operon including *cj0411*, *cj0412* and *cj0413* (4710 base pairs) were amplified by PCR and inserted into the MCS site of pUC19 to create pUC19_cj0411-13. Using PCR and Gibson assembly, base pairs 300–3980 from the 5′ end of *cj0411*, *cj0412* and *cj0413* operon were excised from pUC19_cj0411–13 and replaced by a kanamycin resistance cassette. Within the operon, *cj0413* initiates at base pair 4006 meaning this gene is unaffected by the kanamycin resistance cassette insertion. The resulting plasmid pUC19_Δcj-dlp1/2 was electroporated into wild type *C.jejuni* 81–176[55] and plated onto MH agar containing 50 µg/ml kanamycin. The desired knockout, called Δcj-dlp1/2, was verified by colony PCR and sequencing, and stored in MH media supplemented with 15% glycerol at −80 °C.

**Subcellular fraction assay.** Subcellular fraction of *C. jejuni* was performed as described previously with small modifications[56]. In brief, freshly grown cells in MH media were harvested and washed by PBS. For generating periplasmic and cytoplasmic fractions, cells were incubated with PBS buffer containing 1 mg/ml polymyxin B sulphate and 5 mM EDTA at 4 °C for 1 h. After centrifugation at 4 °C at 16,000 × g for 30 min, the spheroblasts were pelleted. The supernatant was removed and centrifuged again for 30 min at 16,000 × g. The resulting supernatant represents the periplasmic fraction. The spheroblasts were resuspended in PBS and disrupted by Lysing Matrix B (MPbio) on FastPrep-24 5 G Homogenizer (MPbio). After spinning at 12,000 × g for 5 min to remove unbroken cells, the sample was centrifuged at 150,000 × g at 4 °C for 1 h to pellet the membranes. The supernatant was further centrifuged at 150,000 × g at 4 °C for 30 min, resulting in soluble cytoplasmic fraction. For inner membrane and outer membrane fractions, the pellets were incubated with PBS containing 1% N-lauroylsarcosine sodium at room temperature for 1 h. The sample was centrifuged for 1 h at 4 °C at 150,000 × g, resulting in soluble inner membrane and insoluble outer membrane fractions.

Each fraction with equivalent cell numbers were analysed by SDS-PAGE. iBlot dry blotting system (Invitrogen) was used to transfer proteins from SDS-PAGE to membrane. To generate antisera of HcpA, the *cj0413* gene was cloned into pHis17 vector with a C-terminus 6 x His tag, expressed and purified through refolding as described[42]. Antisera against Cj-DLP1, Cj-DLP2 and HcpA (*cj0413*) were raised in rabbit respectively (Cambridge Research Biochemicals). Antisera against RpoA, MotA and FlgP *Campylobacter jejuni* were generous gifts from David Hendrixson, UT Southwestern Medical Center. The α-Cj-DLP1, α-Cj-DLP2, α-HcpA, α-RpoA, α-MotA, α-FlgP polyclonal antisera were diluted 1:2000, 1:500, 1:3000, 1:2000, 1:5000, 1:20,000, respectively. After washing, rabbit, mouse, or guinea pig HRP conjugated secondary antibody was used according to manufacture protocol. Blots were developed using Luminata Crescendo Western HRP Substrate (Millipore) and visualised using a ChemiDoc MD station (BioRad).

**Molecular dynamics.** Hetero G-domain dimer model: a candidate Cj-DLP1/2 G-domain heterodimer was created by aligning Cj-DLP1 and Cj-DLP2 onto different chains of the Mitofusin 1 G-domain homodimer (5GOM) using FATCAT. The resulting fit had no overlapping atoms and was relaxed in explicit solvent with CHARMM26 for 10 nanoseconds. The resulting heterodimeric structure was used to define the attractive interface contacts between G-domain heterodimers via the Shadow definition[57] to be used in the structure-based potential.

Structure-based potential: The structure-based model (SBM) is a protein simulation potential energy function based on the energy landscape theory of protein folding[58]. Here, the SBM was generated using the SMOG v2.0.3 software package (http://smog-server.org)[59] with the 'SBM_Calpha+gaussian' forcefield[60]. The native structure was Cj-DLP1/2$_{tetramer}$ with GDP·AlF$_4^-$. Cj-DLP1 trunk domains were removed and a linking loop added between M463 and N705 using SWISS-MODEL. The crystal contact between Cj-DLP1$_{AD}$ and Cj-DLP2 at its neck and trunk interface (Supplementary Fig. 4a) was removed. The protein was represented as a single bead per residue at the position of the Calpha carbon. Stable heterodimerisation between G-domains was made possible by additionally including the interface contacts determined from the hetero G-domain dimer model as short-range attractive interactions. Making them half as strong as the native interactions populated the heterodimerised configuration 50% of the time during the simulation.

Molecular dynamics (MD) simulations: The accessibility of heterodimerisation was determined through MD simulations of the SBM. The simulations were performed with Gromacs v4.5.3 modified to include the Gaussian contact potential (available at http://smog-server.org). Simulation was performed using reduced units, a time step of 0.0005, and a Langevin thermostat with a coupling constant of 1. The reduced temperature was 0.92, a temperature low enough to prevent unfolding of the individual proteins, but high enough to sample reversible heterodimerisation. The trajectory used to create the free energy surface in Fig. 5d corresponds to $5 \times 10^9$ time steps. In addition $5 \times 10^7$ time steps are detailed in Supplementary Fig. 4b. The reaction coordinate is the distance between the centres of mass of the respective GDP molecules, which is <2 nm in the docked heterodimer. Also monitored is the fraction of heterodimeric contacts included as stabilising interactions that are within 120% of their distance in the modelled Cj-DLP1/2 heterodimer.

**Data availability**. The atomic coordinates have been deposited in the Protein Data Bank (PDB) under accession codes 5OWV (Cj-DLP1/2$_{tetramer}$ apo) and 5OXF (Cj-DLP1/2$_{tetramer}$ GDP·AlF$_4^-$).

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

## Acknowledgements

We thank the beamline staff at Diamond synchrotron. Tillmann Pape and Marc Morgan for facility support. We acknowledge the gift of *Campylobacter* antibodies from David

Hendrixson. Morgan Beeby, Bonnie Chaban, Brendan Wren and Ozan Gundogdu gave *Campylobacter* materials and expertise. Jan Löwe and Frank Bürmann gave manuscript feedback. This work was supported by a Wellcome Trust Fellowship (097328/Z/11/Z) to H.L.

## Author contributions

J.L. and H.L. designed experiments. H.L. initially purified proteins and obtained crystals. J.L. purified proteins, significantly optimised crystals, collected data and solved structure with contributions from H.L. J.L. built and refined the structure, and performed all other experiments except MD simulations which were performed by J.N. H.L. wrote the paper with contributions from J.L. and J.N.

## Additional information

**Competing interests:** The authors declare no competing interests.

