## [Peer Review File · Nature Communications]

Reviewers' comments:

Reviewer #1 (Remarks to the Author):

This manuscript from Harry Low and co-worker describes the structure of a bacterial dynamin pair composed of the *Campylobacter jejuni* Cj-DLP1 and Cj-DLP2. The authors provide an atomic model of a stoichiometric tetramer that is made up by two heterodimers (Cj-DLP1 and Cj-DLP2). The apo form of the tetramer shows a remarkable and unexpected conformation with the two Cj-DLP2 molecules in the center forming a "back-to-back cis-dimer" flanked by a DLP1 part on either side.

This work is elegantly presented, easy to follow and adds a very important piece of information to our knowledge of how DLPs tether opposing membranes. The elucidated structure is able to explain membrane tethering in trans and is of interest not only for microbiologists, but for the entire community working on membrane dynamics in eu- and prokaryotes. There is really little to criticize on this manuscript and I would like to congratulate the authors to their work. Since the first structure of a bacterial dynamin published by Low and Lowe it was unclear how the hetero-oligomeric DLPs exert their function. This work solves now a couple of important questions.

I have only minor points for a revision. There are few things that have to be changed (first points) and a few that may make the ms even stronger, but this reviewer does not consider them essential.

Must change in revision:

1. The Figure S1 is not correct!
2. In Figure S3 I would like to see the overall (holo)-structure with nucleotide (not only the cropped DLP1 and 2 parts). Even if the resolution is not so high, the relative positioning of the domains should be visible.
3. Page 14, line 10: Reference to DynA nucleotide independent tethering is missing.
4. Page 14, line 18: DynA from *B. subtilis* was also shown to be inactive when either the DLP1 or DLP2 part were mutated, or when these parts were analyzed separate.
5. Page 14, line 25: DynA was described to localize mainly at the inner side of the plasma membrane – also in the absence of nisin. In presence of nisin it was described as focal points.
6. Page 15, line 9: This sentence sounds as if the epsilon proteobacterium *C. jejuni* would sporulate. I guess they refer to *Streptomyces* here. If at all, Cj-DLP1 and Cj-DLP2 may influence vegetative division. However, the authors have a null allele and did not describe any growth phenotype (and the EM image in suppl. Figure 6 does not suggest a division phenotype either). Could the authors therefore exclude a role in cytokinesis in *C. jejuni*?
7. Figure S7. In the last step of the model, it suggests that GTP binds and GDP leaves the molecule. I guess this is not formally correct.
8. Page 16, line 8-9. I agree that the experiments suggest membrane fusion rather fission. This would be in accord with the data published for the *Bacillus* DynA where at least lipid mixing was shown. A model has been proposed in a review: *Biol Chem.* 2012 393(11):1203-14. doi: 10.1515/hsz-2012-0185.

9. Could be added in a revision

The authors show nicely the importance of the H1 connector. A mutant lacking this H1 helix would be nice to see and I wonder whether the authors may even have looked at such a mutant. Would this result in a loss of functional interaction between DLP1 and 2?

Reviewer #2 (Remarks to the Author):

In the presented manuscript Liu and colleagues describe the tetrameric nucleotide-free and GDP.AIFx-bound crystal structures of a complex of two full-length dynamin-like proteins from the pathogenic bacterium *Campylobacter jejuni* termed Cj-DLP1 and Cj-DLP2. Cj-DLP2 forms the core of the complex in the form of a parallel dimer, whereas one Cj-DLP1 monomer binds to one Cj-DLP2 monomer each. One of the conformations observed for Cj-DLP1 provides the basis for tethering two opposing membranes for subsequent fusion. Similar heterotypic complexes have also been postulated as functional units for human mitofusins 1 and 2 implicated in mitochondrial membrane fusion events, although no such structure is available to date. Combining their structural analysis with a wide panel of mutational, biophysical, and computational studies, the authors convincingly argue that the two proteins function in the tethering of distant membranes to achieve membrane fusion. Given the topological similarities between the bacterial DLPs and the mitofusins, the mechanistic conclusions drawn from this study are likely to be relevant for the latter.

The paper is well-written and altogether provides sufficient data to support the arguments of the authors. I find the paper suitable for publication in Nature Communications after several issues have been addressed.

Major points

1. The authors neither mention the autoinhibition that must exist to maintain Cj-DLP2 in the monomeric state, nor how it is released upon binding of Cj-DLP1. Although in the absence of a monomeric structure such considerations must remain speculative, an appropriate section in the discussion may still be interesting.
2. Supplementary Figure 1 has not been included in the manuscript, but has been replaced by a copy of Figure 1. Thus, the content of Supplementary Figure 1 could not be assessed.
3. The recent report of the mitofusin 1 dimer structure by Yan et al. (NSMB, 2018) should be cited and included into the discussion as appropriate.

Minor points:

4. (page 11, line 18 and axis inscription of Figure 5e): The designation 3.6 $\mu\text{M}/\text{min}$ seems somewhat incomplete. To which amount of protein does this refer to?
5. (page 13, line 20): Overall, these data are ...
6. (page 32, Fig. 1d+e): The abbreviation for the molecular mass should read kDa instead of KDa.
7. (page 37, lines 8-11 of figure legend): In the text, the capital letters for the inset in Fig 5d (A-D) are written as small letters, whereas the small letters describing the individual figure parts are in capitals.

8. Some of the references are not properly formatted within the text. See page 13, line 19; page 16, lines 13 and 21.

Structural basis for membrane tethering by a bacterial dynamin-like pair

Authors: Jiwei Liu, Jeffrey K. Noel, Harry H. Low

We are pleased to hear that the reviewers were overall positive about our work and would like to thank them both for giving their time to produce insightful and constructive comments. The manuscript has benefited from these revisions.

Reviewer #1:

1. The Figure S1 is not correct!

Many apologies for this rather embarrassing oversight. We have included the correct Fig S1.

2. In Figure S3 I would like to see the overall (holo)-structure with nucleotide (not only the cropped DLP1 and 2 parts). Even if the resolution is not so high, the relative positioning of the domains should be visible.

Fig S3 has been amended to include the overall holo-structure showing GDP as red spheres.

3. Page 14, line 10: Reference to DynA nucleotide independent tethering is missing.

The reference has been included.

4. Page 14, line 18: DynA from *B. subtilis* was also shown to be inactive when either the DLP1 or DLP2 part were mutated, or when these parts were analyzed separate.

An additional line has been included in the text after P14 L21 reflecting this useful point.

5. Page 14, line 25: DynA was described to localize mainly at the inner side of the plasma membrane – also in the absence of nisin. In presence of nisin it was described as focal points.

Thank you for clarifying this. P14 L23 has been adjusted to include this point and some of the following paragraph minorly reworked so that it reads properly.

6. Page 15, line 9: This sentence sounds as if the epsilon proteobacterium *C. jejuni* would sporulate. I guess they refer to *Streptomyces* here. If at all, Cj-DLP1 and Cj-DLP2 may influence vegetative division. However, the authors have a null allele and did not describe any growth phenotype (and the EM image in suppl. Figure 6 does not suggest a division phenotype either). Could the authors therefore exclude a role in cytokinesis in *C. jejuni*?

Agreed this sentence did not read appropriately and indeed we were referring to *Streptomyces*. The reviewer insightfully suggests that a role in cytokinesis may tentatively be excluded given the null mutant, which we had not thought about. We have amended the text to acknowledge this point P15 L11-14.

7. Figure S7. In the last step of the model, it suggests that GTP binds and GDP leaves the molecule. I guess this is not formally correct.

The figure has been amended to show the emission of GDP + Pi

8. Page 16, line 8-9. I agree that the experiments suggest membrane fusion rather fission. This would be in accord with the data published for the *Bacillus* DynA where at least lipid mixing was shown. A model has been proposed in a review: *Biol Chem.* 2012 393(11):1203-14. doi: 10.1515/hsz-2012-0185.

A sentence acknowledging this point and including the suggested reference has been added to the text at P16 L13-14.

The authors show nicely the importance of the H1 connector. A mutant lacking this H1 helix would be nice to see and I wonder whether the authors may even have looked at such a mutant. Would this result in a loss of functional interaction between DLP1 and 2?

This is important, and we have indeed generated this mutant which is shown in Fig. 5b. The Cj-DLP2 Δ H1 mutant has the H1 connector removed. The tetramer is abolished and both Cj-DLP1 and Cj-DLP2 now gel filtrate as monomeric species.

Reviewer #2:

Major:

1. The authors neither mention the autoinhibition that must exist to maintain Cj-DLP2 in the monomeric state, nor how it is released upon binding of Cj-DLP1. Although in the absence of a monomeric structure such considerations must remain speculative, an appropriate section in the discussion may still be interesting.

A discussion has been added P18 L10-16.

2. Supplementary Figure 1 has not been included in the manuscript, but has been replaced by a copy of Figure 1. Thus, the content of Supplementary Figure 1 could not be assessed.

Please see Reviewer 1 point 1.

3. The recent report of the mitofusin 1 dimer structure by Yan et al. (NSMB, 2018) should be cited and included into the discussion as appropriate.

This is an important and timely paper that helps to show how bacterial dynamins are important as model systems for the eukaryotic mitofusins. We have referenced the Yan et al. paper on P16 L17, and then included it in an extended discussion P18 L4-10.

Minor:

4. (page 11, line 18 and axis inscription of Figure 5e): The designation 3.6 $\mu\text{M}/\text{min}$ seems somewhat incomplete. To which amount of protein does this refer to?

The relevant protein concentration has been added to the text P11 L17, Figure 5e legend, and is included in the Methods.

5. (page 13, line 20): Overall, these data are ...

Amended

6. (page 32, Fig. 1d+e): The abbreviation for the molecular mass should read kDa instead of KDa.

Amended

7. (page 37, lines 8-11 of figure legend): In the text, the capital letters for the inset in Fig 5d (A-D) are written as small letters, whereas the small letters describing the individual figure parts are in capitals.

Amended

8. Some of the references are not properly formatted within the text. See page 13, line 19; page 16, lines 13 and 21.

Amended